# Current Understanding of Human Enterovirus D68

**DOI:** 10.3390/v11060490

**Published:** 2019-05-29

**Authors:** Jing Sun, Xiao-Yi Hu, Xiao-Fang Yu

**Affiliations:** Cancer Institute (Key Laboratory of Cancer Prevention and Intervention, China National Ministry of Education), School of Medicine, Zhejiang University, Hangzhou 310009, Zhejiang, China; 21718306@zju.edu.cn (J.S.); 21818322@zju.edu.cn (X.-Y.H.)

**Keywords:** Enterovirus D68 (EV-D68), acute flaccid myelitis (AFM), immune antagonism, infection mechanism, genomic characteristics, infection mechanism

## Abstract

Human enterovirus D68 (EV-D68), a member of the species *Enterovirus D* of the *Picornaviridae* family, was first isolated in 1962 in the United States. EV-D68 infection was only infrequently reported until an outbreak occurred in 2014 in the US; since then, it has continued to increase worldwide. EV-D68 infection leads to severe respiratory illness and has recently been reported to be linked to the development of the neurogenic disease known as acute flaccid myelitis (AFM), mostly in children, seriously endangering public health. Hitherto, treatment options for EV-D68 infections were limited to supportive care, and as yet there are no approved, specific antiviral drugs or vaccines. Research on EV-D68 has mainly focused on its epidemiology, and its virologic characteristics and pathogenesis still need to be further explored. Here, we provide an overview of current research on EV-D68, including the genotypes and genetic characteristics of recent epidemics, the mechanism of infection and virus–host interactions, and its relationship to acute flaccid myelitis (AFM), in order to broaden our understanding of the biological features of EV-D68 and provide a basis for the development of effective antiviral agents.

## 1. Introduction

The genus *Enterovirus* of the family *Picornaviridae* comprises many essential pathogens associated with human and mammalian diseases. This genus is classified into four groups: polioviruses, Coxsackie A viruses (CA), Coxsackie B viruses (CB), and echovirus. It contains fifteen species, encompassing four human enteroviruses (A-D), eight animal *Enterovirus* species (E-L), and three rhinoviruses (A-C). Enterovirus D68 (EV-D68) is a member of the *Enterovirus D* group of the *Picornaviridae* family. It was initially isolated and characterized from the throat swab samples taken from four Californian children with pneumonia and bronchiolitis in 1962 [1], giving rise to four distinct virus strains: Fermon, Franklin, Robison, and Rhyne. In recent years, there has been a gradual emergence of new representative strains, such as US/MO/14-18947(MO), US/KY/14-18953(KY), and others. 

According to the statistics of the US National Enterovirus Surveillance System (NESS), only 26 sporadic cases of EV-D68 were reported from 1970 to 2005 [2]. Despite the fact that EV-D68-associated infection was previously considered infrequent, since 2005 it has undergone a worldwide burst of growth in the USA, Asia, and Europe, with an ever-increasing incidence over the past decade and a half [3]. In 2014, the United States experienced the largest upswing in EV-D68 infection: 1153 cases of EV-D68 were diagnosed and associated with severe respiratory symptoms [2]. A number of retrospective studies later also verified that a high prevalence of EV-D68 infection existed in Europe during the same period [4]. The most common clinical symptom of EV-D68 infection is respiratory illness [5]. As a non-polio enterovirus, EV-D68 was found to have a temporal association with a polio-like neurological disorder known as acute flaccid myelitis (AFM), with symptoms such as dysneuria and muscle weakness, and a causal link is being investigated by researchers [6]. This phenomenon suggests a potential for EV-D68 to induce nervous system disease [7], arousing widespread concern among health authorities and the public.

The genome of EV-D68 contains a positive-sense single-stranded RNA of about 7.6 kb. Like other picornaviruses, EV-D68 consists of a single open reading frame (ORF) flanked by untranslated regions (UTR) at each end [8], encoding a precursor polyprotein that is further cleaved autocatalytically to yield four structural proteins (VP1, VP2, VP3, and VP4) and seven non-structural proteins (2A, 2B, 2C, 3A, 3B, 3C, and 3D) (Figure 1) [8]. VP1, which contains the serotype-specific neutralization BC loop site, is used to distinguish among virus serotypes and to detect newly emerging strains [9]. Viral protein 2C has been reported to play an important role in the processes of infection and viral-host interaction, making it a promising target for anti-EV-D68 drugs. The selective serotonin reuptake inhibitor fluoxetine (Prozac) inhibits human enterovirus by targeting protein 2C and interfering with viral RNA replication [10]. However, fluoxetine has proven to be ineffective in improving the neurologic outcomes of EV-D68-associated AFM in a clinical trial [11]. Recently validated as a novel antiviral target, 2A^pro^ is potently inhibited by telaprevir, an FDA-approved drug used for the treatment of hepatitis C virus infection [12]. The 3C protease contributes to the life cycle of viral infection and replication by inhibiting host innate immune responses; thus, it is also a potential target for antiviral drugs [13,14,15]. 

Recent studies on EV-D68 have mainly focused on epidemiology; the lack of basic research on its virologic characteristics and pathogenesis have hindered our further understanding of this virus. In this review, we will discuss advances regarding EV-D68’s genetic characteristics, infection mechanisms, interaction with hosts, and relationships with neurological diseases in order to provide useful information for future research into the pathogenesis and antiviral therapeutics of EV-D68.

## 2. Genetic Characteristics of EV-D68

EV-D68 is mainly divided into four subtypes and many branches, including the original classical subtypes and recent epidemic-related branches: Clade A, Clade B, Clade C, and Clade D (Figure 2). The location and corresponding year of specimen collection of EV-D68 VP1 sequences have revealed that EV-D68 subtype Clade A, Clade B, and Clade C are prevalent in multiple regions of American, Europe, and Asia. Several viral subtypes have often co-circulated, but Clade C is the least common of the subtypes. The major 2014 outbreak and currently prevalent strains in the United States are represented by Clade A1, Clade B1, Clade B4, and Clade B5 subtypes, whereas, also in 2014, European countries, such as Italy and the Netherlands, found confirmed cases of Clade A1, Clade A2(which has been reclassified to Clade D since 2016), Clade B1, and Clade B2 infection. In Canada, the major prevalent strain has been Clade B2. In Asian countries such as Thailand and Japan, the detection of Clade B and Clade C have been reported, whereas the main subtypes in China have been Clade A2 (which has been reclassified to Clade D since 2016) and Clade B2 [16]. The emergence of EV-D68 subclade D1 was recently reported in France and northern Italy, from August to November, 2018 [17,18]. Xiang et al. [18] reported that genogroup replacement has occurred in China since 2006. Before August 2011, Clade A was predominant in EV-D68-infected cases in China, and other strains were only scattered. Clade B emerged in October 2011 and co-circulated with Clade A until November 2013. However, in 2014, the circulating strains shifted from Clade A to B [19]. Genogroup replacement is one of the major causes of new outbreaks; fortunately, the genogroup replacement in China did not incur an EV-D68 upswing [19]. 

Apart from the virus itself, community immunity, and especially the pre-existing virus serum neutralizing antibody (NAbs) in the community, are very likely to influence the spread and prevalence of the virus. Research on viral NAbs of specific populations can trace the history of viral infection and predict the susceptibility of a population to certain viruses. Xiang et al. [20] collected serum samples from individuals with EV-D68 infection in Beijing, China from 2004 to 2011, tested the seroprevalence of EV-D68 NAbs, and found that the level of EV-D68 NAbs in the Chinese community in 2004 was remarkably lower in all age groups than in 2009. Furthermore, from 2007 to 2011, the NAbs against EV-D68 in the population increased over time. Meanwhile, researchers also found that the NAbs actually can inhibit virus replication. Thus, the high level of EV-D68 NAbs in the Chinese community could be one of the reasons why EV-D68 infection in China appears only sporadically instead of in the form of outbreaks.

Recombination and mutations caused by the error-prone polymerase, human immune response, and population polymorphism have been recognized as the main mechanisms for enterovirus evolution [21]. It has been reported that the US EV-D68 outbreak in 2014 was caused by a novel clade [7,22]; researchers detected six kinds of gene mutation, M291T, V341A, T860N, D297N, S1108G, and R2005K in a distinct clade B1 of EV-D68 strains, which were associated with the neurovirulence of acute flaccid myelitis (AFM) in American strains [7,22]. Moreover, through comparative analyses of EV-D68 2014 outbreak isolate sequences, Zhang et al. also identified that AFM/AFP-associated isolates belong to a single phylogenetic subclade, B1, with 12 substitutions that carry the same residues observed at equivalent positions in paralysis-causing enterovirus such as poliovirus, EV-D71, and EV-A71 [23]. Moreover, subclade B3, now the most prevalent EV-D68, is also reported to be associated with AFM [24]. Later, Huang et al. discovered three variants (C187T, C3277A, and A4020G) that differ from those previously identified through comparative genomic analysis. C3277A and A4020G cause amino acid substitutions T860N and S1108G, respectively, which are cleavage sites for the viral proteases 2A and 3C. The mutation of these sites alters the cleavage efficiency, leading to an increased rate of viral transmission and replication [5]. However, there have been cases of infection by EV-D68 strains with neurovirulence-related mutations that have only shown non-neurologic symptoms such as respiratory illness [7]. This result suggests that viral gene polymorphism is necessary to cause nervous system symptoms such as AFM, but other factors also need to be considered. Nevertheless, only one case was reported in China in 2011 in which the individual was infected by the mutant strain that had the six substitutions mentioned above. In 2014, a unique R220A mutation was detected in the EV-D68 strain prevalent in China [18], which might also result in changes in virulence. As in other enteroviruses, the loop sequences of the X and Y domains in the 3’-UTR of EV-D68 are complementary, but Xiang et al. found that the X loop sequences of the US 2014 outbreak EV-D68 strains were not complementary but instead identical to loop sequences of the Y domain, indicating that the structure of the 3’-UTR of the viral genome can affect viral replication. Meanwhile, all the epidemic EV-D68 American strains have C-U, C-A mutations in the 3’-UTR, a finding that may influence the phenotypes of the virus [18]. 

Existing epidemiological evidence indicates that the US has the highest prevalence of EV-D68. However, in recent years, cases of EV-D68 infection in Europe, Asia, and other countries have continued to increase. The evolution of EV-D68 exhibits geographic and temporal diversity, and changes in genotype largely affect the pathogenicity and epidemic characteristics of the virus.

## 3. Infection Mechanisms of EV-D68

Unlike other enteroviruses, which are acid-resistant and replicate in the human gastrointestinal tract, EV-D68 displays acid lability, prefers a lower growth temperature, and multiplies in the respiratory tract; these are also characteristic features of human rhinoviruses. Thus, EV-D68 shares important biological and molecular properties with both the enteroviruses and rhinoviruses [5]. Neuraminidase (NA) can catalyze the hydrolysis of sialic acid. Previous studies have shown that NA treatment reduces 90% of the attachment of EV-D68 to HeLa cells, indicating that sialic acid might be a potential receptor for EV-D68 [25]. Soile et al. discovered that monoclonal antibodies to decay-accelerating factor (DAF) can inhibit the cytopathic effect induced by EV-D68, suggesting that DAF is possibly the medium of EV-D68 infection [26]. DAF is also a critical cofactor for the infection of African green monkey kidney epithelial cells and primary human endothelial cells by hantaviruses [27]. However, Vero cells are non-permissive for EV-D68 infection, suggesting that DAF is not likely to be the receptor for EV-D68. Liu et al. have demonstrated that cell-surface sialic acid supports the infection of susceptible cells with EV-D68. Crystal structures of EV-D68 in complex with sialylated glycan receptor analogs suggest that the virus binds within the “canyon,” a depression on the surface of the virion, and induces a cascade of conformational changes that eject a fatty-acid-like molecule that regulates the stability of the virus, thereby initiating viral uncoating and cell entry [28,29]. Glycan array analysis has shown that EV-D68 strains preferably bind to α2-6-linked sialic acids (α2-6 SAs) as opposed to α2-3 SAs [30].

The existence of sialic acid on the surface of human respiratory tract endothelial cells may be the reason for the effective replication of EV-D68, which leads to infection of the respiratory system. The modification of sialic acid is also critical for EV-D68 infection, and protein glycosylation in the secretory pathway plays an important role in viral invasion. However, apart from sialylated receptors, some strains of EV-D68 are reported to use an alternative nonsialylated receptor [31]. The dependence of US/MO/14-18947 (MO) and US/KY/14-18953(KY), the circulating strains in the 2014 United States EV-D68 epidemic, on sialic acid was decreased during viral infection, and these strains were resistant to neuraminidase treatment, pointing to the existence of another vital protein receptor(s) that might mediate the entry and invasion of EV-D68 into host cells [32].

In 2016, for the first time, our research team reported a protein receptor for EV-D68, the intercellular adhesion molecule 5 (ICAM5/telencephalin) [32]. ICAM 5, with its nine extracellular lg domains, is the largest member of the ICAM subgroup identified so far. ICAM 5 is much more complex than other ICAMs, and its expression is enriched in the mammalian telencephalon of the central nervous system, whereas other ICAMs are mainly distributed in the immune and blood systems, such as in lymphocytes (ICAM-1 and -3), endothelial cells (ICAM-1 and -2), epithelial cells (ICAM-1), and erythrocytes (ICAM-4). As a somatodendrite-specific adhesion molecule, ICAM-5 not only participates in the interactions between the immune and nervous systems but also contributes to neuronal activity, dendrite targeting signals, and cognition [33]. We found that EV-D68 specifically binds to ICAM-5 and depends on it to invade host cells. EV-D68 replication and infection are clearly attenuated when ICAM5 is silenced or soluble ICAM-5 protein fragments are added. In contrast, EV-D68 replication can occur when ICAM5 is exogenously expressed in EV-D68-non-permissive Vero cells.

In the case of the sialic acid-independent strains US/MO/14-18947 (MO) and US/KY/14-18953(KY), ICAM5 is still required as a cellular receptor, and Asn54-linked glycan has proven to be critical to this process [33]. The identification of ICAM5 as an entry receptor for EV-D68 provides a new way to better understand EV-D68 pathogenesis, and ICAM-5-Fc has potential as a novel therapeutic application for viral intervention against disease caused by EV-D68.

## 4. Virus-Host Interactions of EV-D68

### 4.1. EV-D68 Suppresses the Innate Immune Response

During viral infection, host cells develop various strategies to prevent viral invasion, but viruses have also evolved complex mechanisms to escape and antagonize this host defense. To some extent, the interaction between host and virus is an important factor in determining the course of an infection. The innate immune system is critical to the host’s resistance to viral infection, acting as the first-line physical and chemical barrier to infectious pathogens. When host cells are challenged by pathogenic microbes, different pattern-recognition receptors (PRRs) expressed by the host cells can recognize viral components through the pathogen-associated molecular patterns (PAMPs) of viruses or other pathogens and subsequently trigger intracellular signaling cascades to activate proinflammatory responses [34,35]. These responses lead to the secretion of interferons (IFN) via autocrine and paracrine pathways, stimulating IFN-stimulated genes (ISGs) that can induce an antiviral state in infected host cells [36]. Enteroviruses are mainly recognized by three PRRs: Toll-like receptors (TLRs), retinoic acid-inducible gene I-like receptors (RLRs), and Nod-like receptors (NLRs) [34,35]. A TLR3-TRIF mediated signaling pathway recruits downstream TRAF3 and interacts with IKK-related kinase TBK1/IKKε complexes, resulting in the phosphorylation of IFN regulatory factor 3 (IRF-3). Phosphorylated IRF-3 dimerizes and is translocated to the nucleus, where it induces the production of type I IFN [37]. At the same time, TRIF also recruits TRAF6 and RIP1 to activate TAK1 and IKK complexes (IKKα, IKKβ, NEMO), which mediate NF-κB activation and the subsequent expression of various inflammatory cytokine genes [37,38]. Non-structural viral proteins (2A, 2C, 3A, and 3C) of many enteroviruses, such as EV-A71, CV-B3, and CV-A16, are key players in the innate antiviral response, suppressing interferon production to gain a replication advantage [39,40,41,42,43].

Xiang et al. [12] have reported that 3C(pro) cleaves TRIF at its amino- (Q312) and carboxyl-terminal (Q653) domains, leading to a loss of TRIF’s capacity to activate NF-κB and IFN-β signaling and thus helping enteroviruses to subvert host immune responses. These investigators also found that EV-D68 inhibits host viral immunity by downregulating interferon regulatory factor 7 (IRF7). Furthermore, the process strictly requires 3C(pro) to cleave Q167 and Q189 within the constitutive activation domain of IRF7, preventing IRF7 from activating type I interferon expression to limit viral replication [13]. In our recent studies, we have discovered that 3C(pro) from EV-D68 specifically binds the amino terminus of melanoma differentiation-associated gene 5 (MAD5) and inhibits its interplay with MAVS, consequently blocking MDA5-triggered IFN signaling in the (RLR) pathway [14]. This function is not related to its proteinase activity. On the other hand, 3C(pro) also mediates the cleavage of transforming growth factor β-activated kinase 1(TAK1), leading to the suppression of NF-κB activation, a host response crucial for toll-like receptor (TLR)-mediated signaling. The latest research shows that 2A (pro) acts as a “security protein” of enteroviruses by strongly inhibiting stress granule (SG) formation and IFN-β gene transcription in all human EV species (EV-A to EV-D) [44]. Collectively, 3C(pro) and 2A(pro) of EV-D68 are important molecules in the evasion strategies of EV-D68, counteracting the host’s antiviral response and protecting viral replication (Figure 3). Understanding the molecular mechanisms involved may help us to develop countermeasures to control virus infection. 

### 4.2. EV-D68 Manipulates Cell Cycle Progression

EV-D68 has the ability to manipulate the host cell cycle progression, thus promoting viral production. Although the host cell cycle does not affect viral entry, studies suggest that the G0/G1 phase, but not S phase, promotes viral production, whereas the G2/M phase inhibits viral production [45]. 

The expression of cyclins and CDKs is regulated by EV-D68 to manipulate the cell cycle: On the one hand, cyclin D, CDK4, CDK6, cyclin E1, and CDK2 are downregulated after EV-D68 infection to inhibit the transition from the G0/G1 to S phase; on the other hand, cyclin B1 and CDK1 are downregulated by EV-D68, promoting the transition from the G2/M to G0/G1 phase. All these effects establish favorable conditions for virus production. Furthermore, EV-D68 exerts its host cell cycle regulatory function via its non-structural proteins 3C and 3D. 3C stimulates host cells to leave the G2/M phase by decreasing the expression of CDK1 and cyclin B1; furthermore, 3D increases the percentage of cells in the G0/G1 phase by downregulating cyclin D, CDK4, and cyclin E1. Thus, non-structural proteins 3C and 3D function coordinately to promote viral production and infection [45].

### 4.3. Development of Antiviral Strategies

Currently, there are no effective antiviral drugs or vaccines that are available to treat EV-D68 infection. Treatment options are limited to supportive care for mild and severe cases. The Centers for Disease Control and Prevention (CDC) in the United States have tested 16 drugs as possible candidates for EV therapy. Among them, V-7074, being developed to combine with pocapavir to treat polio infection, and DAS 181, being developed for the treatment of influenza and parainfluenza infections, are thought to have the potential to inhibit EV-D68 viruses [46]. Other potent compounds like rupintrivir, enviroxime, and others, are under pre-clinical evaluation [47]. 

The design and development of effective antiviral therapeutics rely on virology and immunology understanding of EV-D68. The crystal structure of it was revealed by Liu et al., and they also showed that pleconaril, the capsid binding anti-rhinovirus compound, has the ability to inhibit EV-D68 at a half-maximal effective concentration (EC_50_) of 430nM, therefore making it a potential drug candidate for EV-D68 [29]. Just recently, Zheng et al. [48] reported the cryogenic electron microscopy (cryoEM) structures of three EV-D68 capsid states, the procapsid, mature virion, and A-particle, which represent the major phases of the virus. They further described two original monoclonal antibodies (mAbs), 15C5 and 11G1, that inhibit EV-D68 infection with distinct binding properties and neutralizing mechanisms. The A-particle is selectively bound by 11G1, whereas 15C5 binds and triggers mature virions to transform into classical A-particles, mimicking host receptor interactions and thereby neutralizing viral infection. Both mAbs exhibit protective efficacy in vivo. Moreover, mAb A6-1, isolated from an EV-D68-infected Rhesus macaque (*Macaca mulatta*), was found to bind to the DE loop of EV-D68 VP1 and interfere with the interaction between EV-D68 andα2,6-linked sialic acids of the host cells; it can also inhibit EV-D68 infection in mice [49]. The illustration of EV-D68 structure and the discovery of mAbs provide for a structure-based approach to the development of EV-D68 vaccines and therapeutic interventions and also provide a tool for further investigating the biological features and virus–host interaction mechanisms of EV-D68. 

## 5. EV-D68 and Acute Flaccid Myelitis (AFM)

### 5.1. Epidemic Evidence of the Association between EV-D68 and AFM

Except for its respiratory and febrile symptoms, EV-D68 infection is reported to have an association with acute flaccid paralysis (AFP), a neurological disorder. In some cases, it can produce symptoms of acute limb weakness, even paralysis, with distinctive abnormalities of the central spinal cord apparent in magnetic resonance imaging [50]. All of these effects identify EV-D68 as an emerging health-threatening pathogen. In 2012, a total of 23 cases of AFP (mostly children and young adults) in California were reported to the US Centers for Disease Control and Prevention (CDC). EV-D68 was detected in the respiratory specimens of 2 of the 19 available specimens [51]. In the summer and fall of 2014 and 2016, EV-D68 circulated and caused a nationwide viral outbreak in the United States, and the number of cases of AFP also increased significantly during this period. In contrast, the occurrence of AFM in 2016, when EV-D68 was not prevalent, was relatively rare [52,53,54]. 

According to surveillance by the CDC, between August and December, 2014, coincident with the national outbreak of EV-D68, 120 cases of AFM were diagnosed. The incidence of AFM in California grew from baseline (0.028 cases) to 0.16 cases per 100,000 people each year, and EV-D68 was the most common virus detected in the upper respiratory tract specimens from individuals with AFM [55]. From July through October, 2016, the number of confirmed AFM cases again went up dramatically: 144 individuals were confirmed to have AFM [29], accompanied by increases in circulating respiratory EV-D68 [56]. EV-D68 is still the main pathogen detected in respiratory specimens from AFM patients [48,49]. In contrast, in 2015, when EV-D68 was not prevalent, only 22 sporadic cases of AFM were reported [29]. 

In 2016, EV-D68-associated AFM cases also rose notably in Europe [57]. During 2018, the United Kingdom experienced an increase in reported AFP cases that coincided temporally with an upsurge in EV-D68 activity; EV-D68 was detected in the respiratory tracts of 37.5% of the diagnosed AFM cases [58]. Moreover, as compared to 2017 (35 cases), the number of confirmed AFM cases (223) in 2018 in the US is significantly higher and has been accompanied by an increase in EV-D68 infection [59]. The temporal and geographically link between the spread of EV-D68 and the clustering of AFM suggests a strong epidemiological correlation between EV-D68 and AFM.

Studies indicate that it is difficult to discover any pathogens, including EV-D68, in the cerebrospinal fluid (CSF) of AFM patients. This lack of direct evidence for a central nervous system etiology presents a major obstacle to further investigating the relationship between AFM pathogenesis and EV-D68 infection. Since 2014, EV-D68 has been detected in the spinal fluid of 4 AFM cases out of 558 confirmed cases in the USA, according to CDC [53]. However, there have been a few cases in which EV-D68 was detected in the cerebrospinal fluid (CSF): In 2014, during an EV-D68 outbreak, an American child’s CSF was positive for EV-D68 [48]. EV-D68 was also identified in a child’s CSF specimen in Argentina when a cluster of AFM cases were consecutively reported in 2016 [60]. Moreover, the autopsy of an individual who died of meningomyeloencephalitis and severe acute mixed pneumonia, with symptoms of myalgia and progressive arm weakness showed EV-D68 infection of the CSF and a characteristic histopathological pattern of enteroviral central nervous system infection [61]. Genomic analysis of the nasopharyngeal (NP) swabs and CSF material from a cluster of 11 pediatric suspected cases of AFM in Phoenix, Arizona in September 2016 also revealed that EV-D68 was present in NP swabs of 3 of 4 confirmed cases. However, no EV-D68 was detected in the CSF samples from these patients. [62]. Moreover, subclade B3, now the most prevalent EV-D68, is reported to be detected in the bronchoalveolar lavage fluid and NP aspirate of an AFM pediatric patient [24]. This evidence further strengthens the likelihood that EV-D68 is the etiologic agent of AFM in these cases and providing reliable biological evidence for further study of this association.

### 5.2. Neuropathogenicity of EV-D68 In Vitro and In Vivo

Brown et al. reported that a subset of contemporary EV-D68 strains, including isolates from the 2014 outbreak, could infect and replicate in neuroblastoma-derived neuronal cell line SH-SY5Y, and neurotropism was also observed in the primary human neuron cultures and a mouse paralysis model [63]. By constructing chimeras at the capsid-level of EV-D68, Royston et al. found it exhibit tropism in respiratory, intestinal, and neural tissues [64]. These in vitro results indicate the EV-D68 neural tropism and signatures of neurovirulence biologically, which provide useful information for the pathogenesis of EV-D68 and its possible associations with AFM.

Four strains from the 2014 outbreak, US/MO/14-18947(MO), US/KY/14-18953(KY), US/IL/14-18952(IL), and US/CA/14-4232(CA), were found to be able to induce a paralytic disease in mice resembling human AFM, whereas EV-D68 prototype strains Fermon and Rhyne could not [65]. EV-D68 infection was able to cause a loss of spinal cord motor neurons, and a virus isolated from the spinal cords of infected mice could transmit the disease to naive mice. Neutralizing antibodies were capable of reducing the paralysis in the mice, and immunosuppressants such as dexamethasone worsened the motor impairment and increased mortality [65,66]. In addition, EV-D68 exhibited strong tropism to the muscles, spinal cord, and lung in a neonatal mouse model and produced limb paralysis, tachypnea, and even death [67]. These in vivo experiments in mice verified the ability of EV-D68 to induce disorders in the nervous system and offered experimental evidence to support research into the relationship between EV-D68 and AFM. Furthermore, our research team found that neuron-expressed ICAM-5/telencephalin is a cellular receptor that is needed for EV-D68 to infect and enter host cells [32]. ICAM-5 is solely expressed in the soma and dendrites of neurons of the mammalian telencephalon, including cranial nerves I and II [68]. Messacar et al. [6] have noted that EV-D68 can also infiltrate the central nervous system through functional receptors located at the end of the olfactory nerve in the upper nasal cavity epithelium. The ability of EV-D68 to infect the host through neuron-specific molecules may indicate a potential link to neurological diseases and provide clues concerning the possibility of EV-D68 infection-associated AFM.

In summary, many epidemiological studies and a small number of basic science studies have established a close association between EV-D68 infection and AFM pathogenesis [6,7], although more definitive proof is required to confirm their relationship. Therefore, in-depth basic research in virology and immunology is critical and should provide strong support for the effective prevention and treatment of AFM. 

## 6. Conclusions

In recent years, infections with the once rarely reported enterovirus D68 have increased remarkably worldwide, accompanied by severe respiratory illness and neurological complications. Infection cases are mainly diagnosed in children and young adults and have raised widespread concern. Discovery of the cellular receptors, sialic acid, and neuron-specific receptor ICAM-5, has proven essential for uncovering the mechanisms of EV-D68 pathogenesis. Meanwhile, studies of the effects of non-structural proteins on virus–host interactions have revealed that the virus can promote replication by altering the cell cycle and escaping innate immunity. Thus, it appears that the development of targeted therapeutic interventions to treat EV-D68 infection that involve inhibitors of the non-structural proteins 2C, 3C, and 2A will prove worthwhile. In addition, investigation of the association between EV-D68 and AFM is important for further elucidating the clinical features of EV-D68 and revealing its virologic characteristics. However, much is still unknown about EV-D68: Its nosogenesis, the role of many viral proteins in the replication cycle of the virus, a more comprehensive, detailed picture of the interplay between EV-D68 and host immunity, genetic variation in EV-D68, and the potential for antiviral therapeutics all await further investigation.

## Figures and Tables

**Figure 1 viruses-11-00490-f001:**
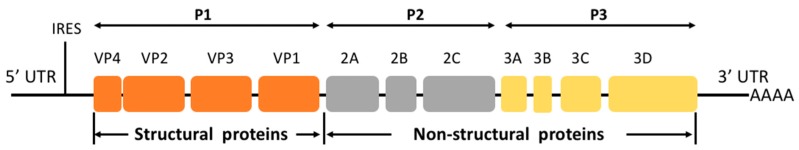
Schematic representation of the enterovirus D68 genome.

**Figure 2 viruses-11-00490-f002:**
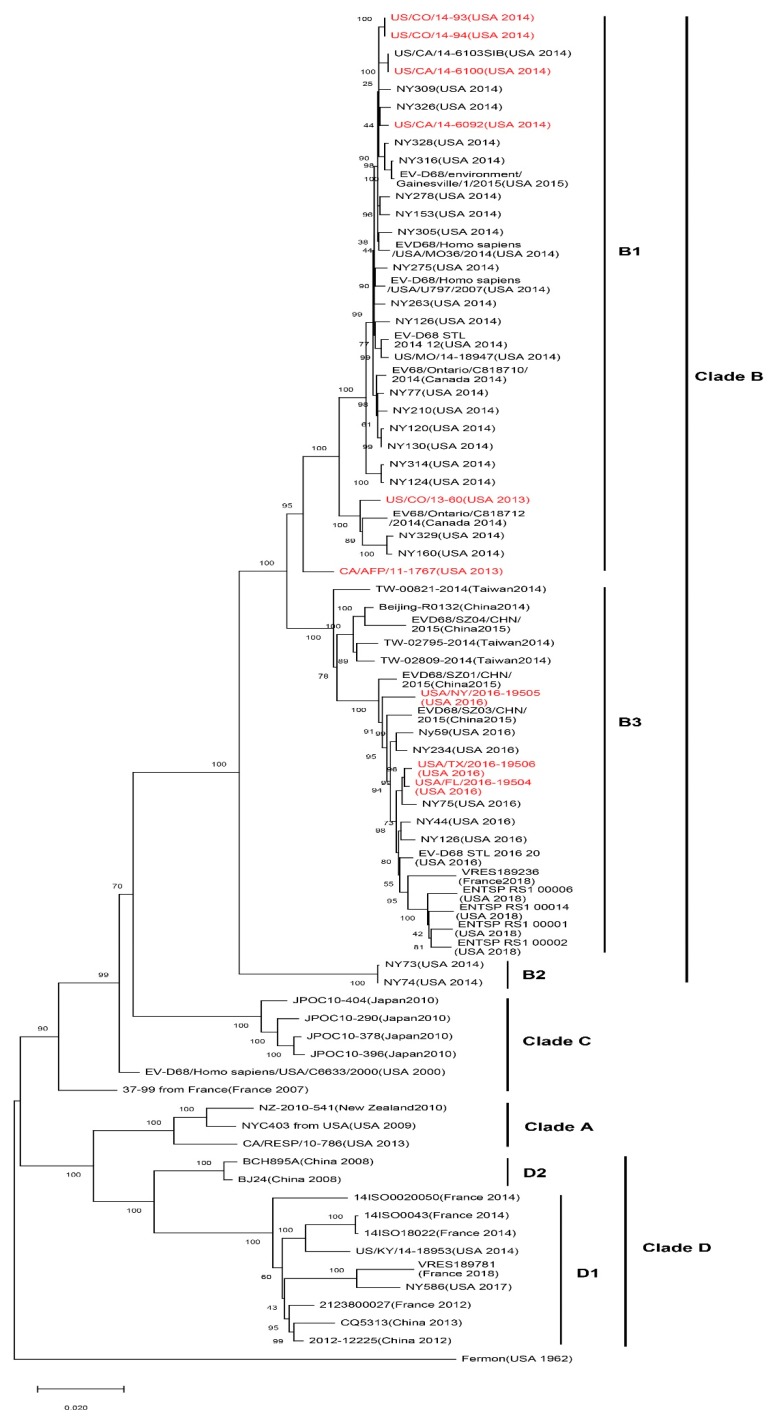
Phylogenetic tree of EV-D68 including acute flaccid myelitis (AFM)-related strains. A maximum likelihood phylogenetic tree of EV-D68 strains based on nearly complete polyprotein gene (1-6388 nt of coding sequences) was constructed using the bootstrap method with 1000 replications based on the Tamura–Nei model in MEGA X. The representative circulated EV-D68 strains during past years were collected in NCBI’s GenBank sequence database (*n* = 76). AFM-related EV-D68 strains are marked in red. AFM-associated stains are mostly in subclade B1 and B3. Branch lengths are drawn proportionally to the number of nucleotide substitutions per position.

**Figure 3 viruses-11-00490-f003:**
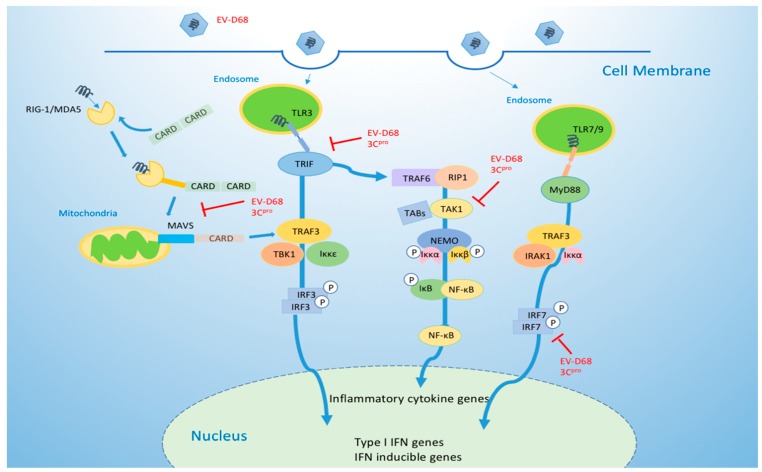
Enterovirus D68 targets important molecules in the toll-like receptor (TLR) and RIG-I like receptor (RLR) pathways to escape innate immunity.

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
