# Peer review of "Current Understanding of Human Enterovirus D68"

_viruses, 2019, doi:10.3390/v11060490_

Round 1
Reviewer 1 Report
The authors have conducted a nice review of this emerging virus, covering many aspects. As outlined below, I suggest updating the epidemiology with some more information on activity in 2018 which was a year of increased activity in North America, and providing some information on antiviral therapies besides monoclonal antibodies.
General comment on nomenclature. As per guidelines at ICTV guidelines, I suggest italicizing genus “Enterovirus”, “Enterovirus D” and “Picornaviridae” (see https://talk.ictvonline.org/information/w/faq/386/how-to-write-a-virus-name)
Specific Comments
Introduction.
Page 1. Line 28 to 31. The ICTV website lists Enterovirus species A to L. Also, poliomyelitis is a member of Enterovirus species C. I suggest making this text a little clearer.
Page 2. Line 44 to 45. The authors state that “EV-D68 was found to cause a polio-like neurological disorder known as acute flaccid myelitis (AFM)”. Note that the US CDC currently is not stating that there is an established causal link as they were only able to detect the virus in 4/558 AFM cases since 2014. See https://www.cdc.gov/acute-flaccid-myelitis/afm-surveillance.html . I would suggest altering the wording to state there has been a temporal association and a causal link is being investigated by health authorities, with some concluding there is sufficient evidence for a causal association.
Ref. 60 by Messacar et al (Lancet Infect Dis. 2018 Aug; 18(8):e239-e247) does evaluate the data for causality, and concludes there is a causal link between EV-D68 and AFM, so could be referred to here when making this statement.
Page 2. 2. Genetic Characteristics of EV-D68.
The authors mention that most Canadian strains are A2. However, the Canadian reference used by the authors (Ref 15 https://www.ncbi.nlm.nih.gov/pmc/articles/PMC5331033/) conducted whole genome sequencing on 34 EV-D68 strains from 2014, and found that 33 (97%) were Clade B, most of which were B2 (31/34; 91%). This should therefore be corrected.
Page 3. Lines 86 to 88. The authors state that genogroup replacement is one of the major causes of new outbreaks. It would be useful if a reference can be provided to support this statement.
Page 7. Section 4.3 Development of antiviral strategies
One gap is that the authors do not provide any information about antiviral medications that have been developed or evaluated for activity against EV-D68. I suggest adding a section/paragraph with some information on this to complete the review.
Page 8. Epidemic evidence of association of EV-D68 and AFM
This section should be updated with a mention of increased activity of EV-D68 in the USA in 2018, along with an increase in AFM cases. See this recent publication from March 2019: https://www.cdc.gov/mmwr/volumes/68/wr/pdfs/mm6812a1-H.pdf
The authors should also add CDCs information that EVD68 has only been detected in in the spinal fluid of four AFM cases out of 558 confirmed cases since 2014 https://www.cdc.gov/acute-flaccid-myelitis/afm-surveillance.html
Page 8. Lines 282 to 284. In describing reference 55, the information provided to the reader should be made more accurate. The study included 11 suspected AFM cases, of which only 4 were confirmed AFM cases. Among the 4 confirmed AFM cases, EVD68 was detected in NP swabs of 3 of them. No CSF specimens were EVD-68 positive in this study.
Author Response
We would like to thank the reviewers for their thorough and insightful comments, which have shown us how to significantly improve this manuscript. To the best of our ability, we have tried to address and solve all issues of concern. The enclosed file of revisions have been incorporated in response to the comments.

Reviewer 2 Report
The manuscript entitled "Current understanding of human enterovirus D68" is well written and describes an interesting basic-virology topic, differently from other review on this topic (EV-D68). I have only a major comments regarding the phylogenetic tree presented in figure 2. All the clades and sub-clades should be included in a more updated phylogenetic tree. For instance no clade D sequence were included and based on time reconstruction of EV-D68 evolution, EV-D68 B3 was the most circulated virus in 2016. Thus, the Figure legend should be updated at 2019 EV-D68 strains circulation. In order to better show the relationship between the strains a maximum likelihood phylogenetic tree could be more informative than NJ-tree. Please specify the genomic region used for the analysis. Also only phylogenetic tree including VP1 sequences should be adequately informative.
Author Response
We would like to thank the reviewers for their thorough and insightful comments, which have shown us how to significantly improve this manuscript. To the best of our ability, we have tried to address and solve all issues of concern. The enclosed file of revisions have been incorporated in response to the comments:

Reviewer 3 Report
This review by Sun et al summarizes fundamental biology of human enterovirus D68 (EV-D68). The review is interesting and well-written, however there are some inconsistencies with current literature in paragraph “2. Genetic characteristics of EV-D68”.
Since 2016, sequence analysis has led to the reclassification of the clade A2 into clade D (Gong, 2016, PMID: 27495059). This is important because contemporary D1 strains have acquired neurovirulence, while clade A viruses had not (Brown, 2018, PMID: 30327438). In Fig 2, strains referred to as A2 are actually D1, except BCH895A which belongs to clade D2. This should be corrected in the text and figures.
The paragraph starting line 100 is not very clear. It should include in which clades the mutations were identified as viral background matters. The authors reports 6 mutations associated with AFM while Zhang et al identified 12 substitutions in B1 2014 isolates as associated with AFM. In addition, the authors should cite papers showing that B3 viruses (which are now the most prevalent EV-D68) can cause AFM.
Fig 2 requires several revisions:
- It should indicate which year were the viruses isolated
- There are some missing data in the legend: which sequence were used to build the tree (whole genome sequences only)?, and how were the 60 sequences chosen? They are not representative of the genetic diversity of EV-D68 as there are now much more sequences in B3 clade than B1.
- In the title “novel clade B1 identified after the 2014 US outbreak” is false as B1 was identified before 2014 as revealed by retrospective analysis (see the review of Gong, 2016, PMID: 27495059; or Kramer, 2018, PMID : 30229724)
The author may add one or two sentence in paragraph 2 about mechanisms driving EV-D68 evolution (including recombination).
In paragraph 5, the authors should also discuss papers showing in vitro evidence of EV-D68 neurovirulence (such as Brown, 2018, PMID: 30327438; Royston 2018 PMID : 29630666).
Minor comments
Line 34 : consider changing “prevalent” in “representative”
Line 120 : consider adding “highest” in “US has the highest reported prevalence”
Author Response

(The authors gave the same response as above.)
